# A Machine Vision-Based Method for Tea Buds Segmentation and Picking Point Location Used on a Cloud Platform

**Jinzhu Lu** [1,2,*], **Zhiming Yang** [1,2], **Qianqian Sun** [2], **Zongmei Gao** [3] and **Wei Ma** [4]

1   Modern Agricultural Equipment Research Institute, Xihua University, Chengdu 610039, China; yangzhiming@stu.xhu.edu.cn
2   School of Mechanical Engineering, Xihua University, Chengdu 610039, China; 3120210108222@stu.xhu.edu.cn
3   Center for Precision and Automated Agricultural Systems, Department of Biological Systems Engineering, Washington State University, Prosser, WA 99350, USA; zongmei.gao@wsu.edu
4   Chinese Academy of Agriculture Sciences Institute of Urban Agriculture, Chengdu 610213, China; mawei03@caas.cn
*   Correspondence: lujinzhu@mail.xhu.edu.cn

**Abstract:** The segmentation and positioning of tea buds are the basis for intelligent picking robots to pick tea buds accurately. Tea images were collected in a complex environment, and median filtering was carried out to obtain tea bud images with smooth edges. Four semantic segmentation algorithms, U-Net, high-resolution network (HRNet_W18), fast semantic segmentation network (Fast-SCNN), and Deeplabv3+, were selected for processing images. The centroid of the tea buds and the image center of the minimum external rectangle were calculated. The farthest point from the centroid was extracted from the tea stalk orientation, which was the final picking point for tea buds. The experimental results showed that the mean intersection over union (*mIoU*) of HRNet_W18 was 0.81, and for a kernel with a median filter size of 3 × 3, the proportion of abnormal tea buds was only 11.6%. The average prediction accuracy of picking points with different tea stalk orientations was 57%. This study proposed a fresh tea bud segmentation and picking point location method based on a high-resolution network model. In addition, the cloud platform can be used for data sharing and real-time calculation of tea bud coordinates, reducing the computational burden of picking robots.

**Keywords:** tea buds picking; deep learning; image segmentation; cloud platform

## 1. Introduction

As one of the world's leading tea-growing, consuming, and exporting countries, China's annual tea production, as well as sales, continue to rise year on year. As of 2018, the market share of premium and branded teas was already as high as 90%. With the shortage of human labor, manual harvesting is becoming inefficient. However, the current harvesting machines are unable to perfectly harvest tea buds. As a result, developing intelligent tea bud harvesting robots is a serious constraint for the booming tea industry. With the globalization of the tea industry, research on the mechanized harvesting of tea buds has been carried out around the world [1,2].

Researchers for tea recognition used methods of image binarization and segmentation thresholds based on color models. Wu et al. [3] proposed a k-means clustering method based on the Lab color model to distinguish tea buds from the background. Chen et al. [4] performed three-dimensional reconstruction and measurement of tea trees using fringe projection profilometry. Thangavel et al. [5] proposed keyframe Prewitt edge detection and threshold segmentation for tea bud classification. The above methods are mostly based on color factors for segmentation. Since the background color of tea buds is similar to branches, stalks, and weeds, the segmentation effect is poor and the segmentation efficiency is low only by color. Similarly, illumination and occlusion will cause the extracted tea buds

to form holes or lose their shape. Nowadays, many researchers also use deep learning to segment tea buds. In order to obtain more discerning tea bud features, a variety of semantic segmentation models were used for performance comparison (Qian et al. [6]). Chen et al. [7] used the fusion of a single-shot detector (SSD) and an image enhancement algorithm to improve the speed and accuracy of tea detection.

Once tea buds are identified and segmented, the picking points of tea buds should be further extracted to facilitate subsequent 3D positioning and finally realize machine picking of tea buds. At present, there is more research on the tea buds at traditional picking points. Zhang et al. [8] implemented the recognition of tea shoot tips in natural environments based on the color factor method. The focus was on the use of raster projection contouring to achieve tea bud tip height information acquisition. In addition, some scholars combine traditional image processing with deep learning to determine the picking point. Chen et al. [9] used the faster region-convolutional neural network (Faster R-CNN) algorithm for tea image recognition. However, there is a limited amount of such research.

With the application of 5G networks, the concept of cloud platforms and cloud computing gradually comes into view (Tang et al. [10]). Cloud platforms provide non-professional users with an operational human-computer interface to make decisions and share data at any time (Zhao et al. [11]). The cloud platform is also used in agriculture. Zhang et al. [12] used a cloud platform to increase the size of the tomato spectral dataset. Based on the Cloudino-iot and FIWARE platforms, Franco et al. [13] can perform real-time monitoring of temperature and humidity parameters during seed germination. At present, there is a lack of a cloud platform to carry out two-way data transfer applications to the picking robot.

At present, the identification and segmentation of tea buds are mostly based on the traditional method of the green color factor. This method has a poor segmentation effect and is limited to different varieties of tea buds. Since the segmentation effect of tea buds often directly affects the extraction of picking points from tea buds, it is very important to select an algorithm with universality and a good segmentation effect. At the same time, due to the different growth posture and shooting angle of tea buds, the orientation of tea buds is also different in this image. It is necessary to design a method to extract the picking points of tea buds with different orientations. Meanwhile, the current application of the cloud platform in agriculture, mainly as a monitoring and statistical tool, involves one-way transmission of data to users, which does not fully exploit its advantages. Two-way feedback on cloud platform data will become particularly important in the future.

In this study, we investigated HRNnet_W18 deep learning-based learning for tea bud segmentation. In addition to segmenting various tea buds, this algorithm can reduce the influence of the natural environment and lighting. In order to extract the picking points of tea buds, this study takes the centroid of the tea buds image as the starting point and connects the center of the minimum external rectangle of the tea buds image. This line points to the orientation of the tea stalk, and in the opposite direction is the approximate orientation of the tea buds. The design was intended to solve the problem of different tea bud orientations.

## 2. Materials and Methods

### 2.1. Image Preparation and Processing

The tea dataset used in this experiment is milky white tea from the tea gardens located in Mount Emei, Sihuan Province. The shooting height was 30 to 50 cm, and images with different backgrounds were included in various time periods. A smartphone, the OnePlus 8T, was used to collect $3024 \times 4032$ resolution tea bud images. While deep learning training is performed using Lenovo LEGION (CPU model AMD Ryzen 7 4800 H, 2.90 GHz, Kingston memory 16 GB, GPU RTX2060, Samsung disk 512 G).

Since the original image resolution is too large, it contains too many tea buds, which will reduce the training effect of deep learning. Therefore, the samples are cropped uniformly to images with $512 \times 512$ resolution images. Labelme's annotation tool was used in this study to augment the annotation of 200 images. The tea bud data set is shown in Figure 1. Data augmentation was also used, including random horizontal flip, random up-and-down flip, random rotation, random image capture, random image blur, random brightness, random contrast, random saturation, and random hue. Finally, 70% of them are used for training and 20% for verification. After the model training, 10% of the images are selected to test the model.

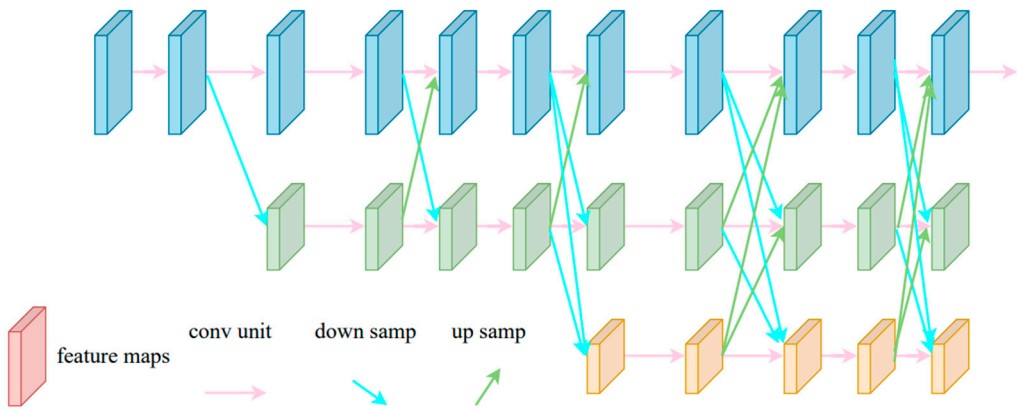

**Figure 1.** Architecture diagram of HRNet used in this study.

### 2.2. Semantic Segmentation

Semantic segmentation is a deep learning algorithm that associates a label or category with every pixel in an image. It is used to recognize a collection of pixels that form distinct categories. In this paper, the most universally used mean intersection over union (*mIoU*) is taken as the index of algorithm effect. *IoU* is the ratio of the intersection and union of each category, while *mIoU* is the average *IoU* of all categories. This index is used for comparison in this paper. Formula (1) below describes the calculation method of *mIoU*, where ($k + 1$) is the number of categories and $P_{ii}$, $P_{ij}$, and $P_{ji}$ represent *TP*, *FN*, and *FP*, respectively (*i* represents a real category and *j* represents other categories). The *IoU* of each category can be regarded as Formula (2).

$$mIoU = \frac{1}{k+1}\sum_{i=0}^{k}\frac{p_{ii}}{\sum_{j=0}^{k}p_{ij} + \sum_{j=0}^{k}p_{ji} - p_{ii}} \tag{1}$$

$$IoU = \frac{TP}{(TP + FN + FP)} \tag{2}$$

In this study, four semantic segmentation algorithms, each with its own characteristics (U-Net, High-Resolution Network (HRNet_W18), Fast Semantic Segmentation Network (Fast-SCNN), and Deeplabv3+), were selected for processing images. U-net has a symmetric network structure and uses its low-level features for feature fusion, which is mainly applied to seamless segmentation tasks. However, this network structure will reduce the resolution of output results. Fast-SCNN works in a multi-branch network structure; branches sharing a shallow network can simplify the whole network structure and reduce the amount of computation. Moreover, the learning to downsample module is designed in this network, which can be used by branches to extract low-level features. Based on the learning to downsample module and two branches, the real-time semantic segmentation network is constructed. Deeplabv3+ proposes a new encoder-decoder architecture that focuses on target boundary information to improve segmentation while using the Xception module

and applying depth-wise separable convolution into the Atrous Spatial Pyramid Pooling (ASPP) and decoder modules to improve the computational speed of the network.

HRNet does not use a cascaded structure such as U-NET, SegNet, DeconvNet, or Hourglass. In order to obtain high resolution for cascaded structures, the general process is to reduce the resolution first and then restore and increase it. Such a continuous up-and-down sampling operation will lose a lot of information. However, HRNet uses a parallel network structure to add interaction between feature maps of different resolutions (Fusion) on the basis of parallel connections. This structural innovation can keep the image at high resolution all the time. The HRNet network structure is shown in Figure 1.

At present, there are seven different open source codes in the HRNet series, among which the Top1 and parameters of HRNet_W64 are 0.793 and 128.06 M, respectively, while the Top1 and parameters of HRNet_W18 are 0.769 and 21.29 M, respectively. Although HRNet_W18's Top1 is 0.024 lower than HRNet_W64's Top1, parameters are 6 times lower. Therefore, choosing HRNet_W18 greatly saves memory space and reduces computing costs.

### 2.3. Realization of Tea Bud Picking Point Extraction Algorithm

After processing the tea bud image, the HRNet_W18 algorithm was used to obtain the tea bud segmentation image. The size selection of the median filter kernel directly affects the effect of picking points. Therefore, five different kernel size selections ($3 \times 3$, $5 \times 5$, $7 \times 7$, $9 \times 9$, $11 \times 11$) were compared in the experiment.

Then, using the center of the minimum external rectangle and the centroid of the tea bud image solved, combined with the corner points calculated by the corner detection algorithm, the tea bud picking points can be obtained. A roughly schematic diagram is shown in Figure 2. The red dot, blue dot, green dot, yellow line, and purple dot in the figure, respectively, represent the detected corner point, centroid, image center, tea stem orientation, and final picking point.

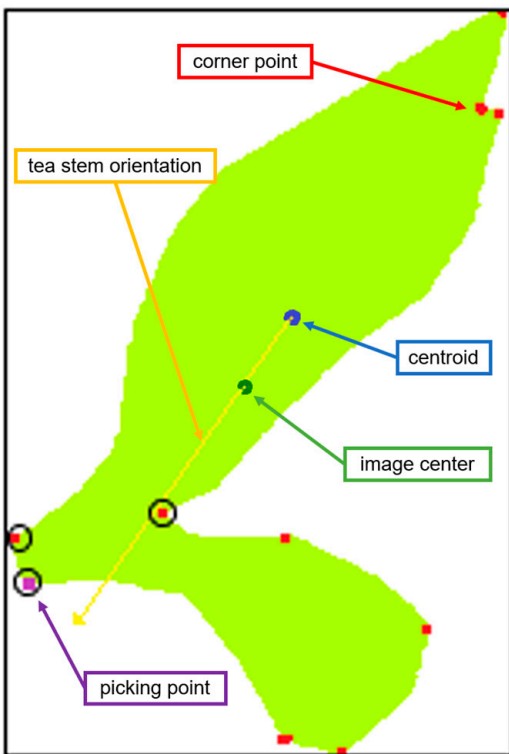

**Figure 2.** Schematic diagram of picking point extraction.

In summary, the segmentation of tea bud images and the specific process of picking point extraction is shown in Figure 3.

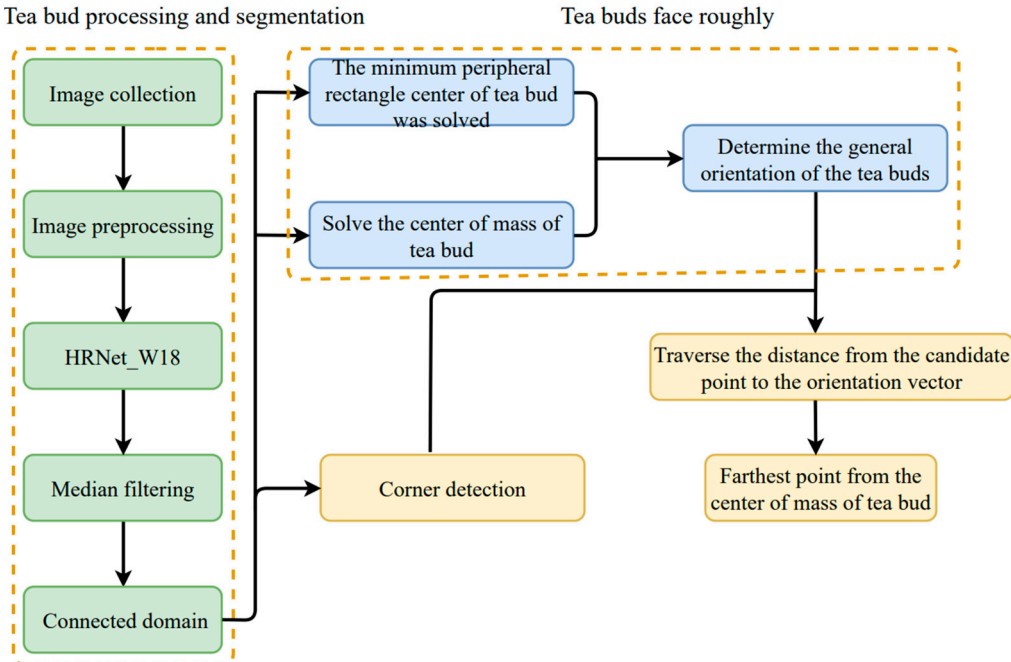

**Figure 3.** The flowchart of tea bud identification, segmentation, and picking point extraction used in this study.

### 2.4. Cloud Platform Architecture

The establishment of a cloud platform can not only include software operations but also enable people to share their data. If you put your server's computing in the cloud, you can replace the machine's mobile processor. At the same time, this will greatly reduce the purchase cost of the machine and the calculation of its energy consumption, increasing its endurance.

The designed cloud platform realizes the process of field image collection, cloud computing, sending back the execution signal, and carrying out the field tea picking action. This system uses the model-view-controller (MVC) three-layer architecture mode to represent business rules, provide an interactive user interface, and accept processing data. The technical framework uses the Mysql database to build the storage center, uses Python 3.7.5 as the platform support, uses the web server gateway interface (WSGI) to realize the communication between web applications and web servers, and ultimately uses the bootstrap framework combined with CSS3 + HTML5 to develop the front-end display interface. The completion of the technical framework layer, middleware layer, data interface layer, and display layer can realize the two-way transfer of cloud platform and robot data.

The cloud platform system consists of two parts: the system architecture and the technical architecture. This system adopts MVC's three-tier architecture mode, and the system architecture diagram is shown in Figure 4 below. M stands for model. The model represents business rules. Among the three components of MVC, the model has the most processing tasks. V stands for view. View refers to the interface that users see and interact with. It is mainly a web page interface composed of HTML elements, which is a way to output data and allow users to operate. C is the controller. The controller accepts the user's input, calls the model and view to fulfill the user's requirements, receives the request, decides which model component to call to handle the request, and finally determines which view to display the returned data.

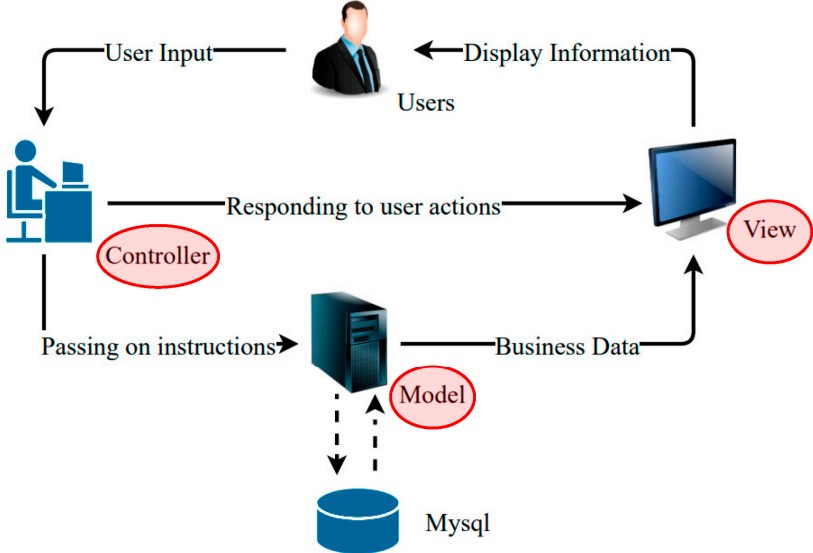

**Figure 4.** System architecture of the cloud platform.

The technical architecture diagram of the cloud platform is shown in Figure 5 below. The technical architecture is divided into the technical framework layer, middleware layer, data interface layer, and presentation layer. The database layer uses the Mysql database to build an efficient data storage center and uses the active replication function of Mysql binary log files to form the Mysql data cluster and establish a Mysql cluster system of "one master and many slaves". On the one hand, real-time hot backup of data is realized; on the other hand, load balancing of data queries and data read and write separation are realized. In the technical framework layer, Python 3.7.5 is used as platform support, Flask is used as the back-end processing framework, and SQLALchemy is used as an ORM. The middleware layer mainly uses WSGI and Apache Tomcat, in which WSGI implements the communication between Web applications and Web servers and Tomcat implements the deployment of background programs as a container. In the front-end display layer, the bootstrap framework is used to develop the front-end display interface in combination with various features of CSS3 + HTML5, which ensure the appearance of the system interface. In addition, due to the complex business logic of the system, the front-end also needs to deal with a lot of business logic and animation effects. Therefore, it uses javascript, jquery, and other front-end technical frameworks and adopts mainstream Ajax technology to ensure data communication between the front and back ends.

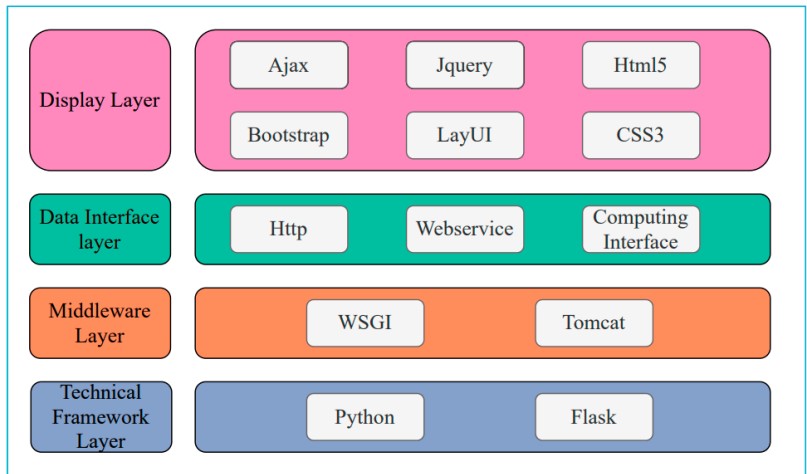

**Figure 5.** Technical architecture of the cloud platform.

## 3. Results and Discussion

### 3.1. mIoU of Semantic Segmentation Models

Four algorithms—Deeplabv3+, U_Net, HRNet_W18, and fast segmentation convolutional neural network (FastSCNN)—are used to segment tea buds. Figure 6 shows the *mIoU* variation diagram of each model with 300 iterations. The best *mIoUs* of Deeplabv3+, U_Net, HRNet_W18, and FastSCNN are 0.7859, 0.7964, 0.81, and 0.748, respectively. The *mIoU* of HRNet_W18 was the largest, and that of FastSCNN was the smallest, which was caused by the decreased resolution of tea buds after sample set cropping. HRNet_W18 can always maintain the state of high resolution in the process of convolution, so it can have a good segmentation effect in the case of low image resolution. In the later shooting sampling, the samples that meet the requirements can be obtained by reducing the depth of field, and the performance of each semantic segmentation model can be improved.

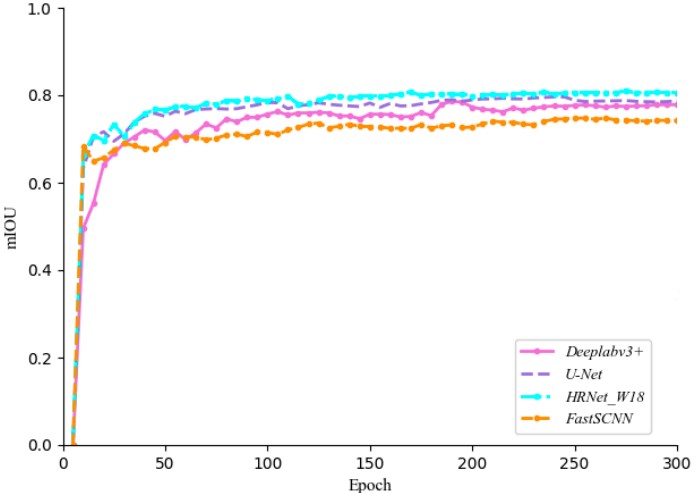

**Figure 6.** *mIoU* of segmentation performance of each model.

### 3.2. Selection Kernel Analysis of Median Filter Kernel

To verify the impact of corner points on media filter kernels of different sizes, 78 images in total of 198 tea bud targets were randomly selected as experimental objects and sent into each nucleus for processing, and the five different kernel sizes were $3 \times 3$, $5 \times 5$, $7 \times 7$, $9 \times 9$, $11 \times 11$. As shown in Figure 7, with the gradual increase in filtering kernel size, the number of tea bud images with abnormal corner points increases. When the $3 \times 3$ filtering kernel size is selected, the abnormal tea bud images have the lowest number at 23. The largest number is $11 \times 11$ kernel, and the number of tea bud images with abnormal corner points is 117, accounting for 59%. When the $3 \times 3$ filtering kernel size is selected, the abnormal tea bud images have the lowest number at 23. The largest number is $11 \times 11$ kernel, and the number of tea bud images with abnormal corner points is 117, accounting for 59%.

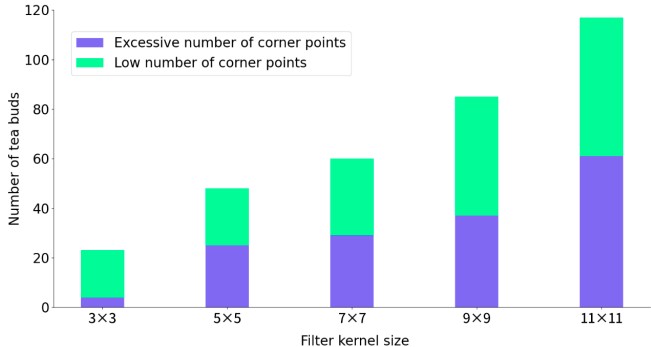

**Figure 7.** The number of tea bud images with abnormal corner numbers under different kernel sizes.

As the filter core moves to the edge of the image, the non-target part of the image will be filled with 0, resulting in the deformation of the edge image covered by the filter core. Therefore, with the increase in size of the filter kernel, the larger the non-target image covered by the filter kernel, the more serious the image edge deformation will be Figure 8 shows two cases of an abnormal number of corners. After HRNet_W18 segmentation and prediction, the image quality of the original tea bud is uneven. Some tea bud images are relatively smooth, and after further use of median filtering, image edges will become smoother, resulting in a low number of corner points, as shown in Figure 8a. Due to the complexity of the environment, the edges of some segmented tea bud images appear jagged. The selection of a $3 \times 3$ filtering kernel for smooth filtering makes it difficult to eliminate the jagged edges of images, resulting in an excessive number of corner points, as shown in Figure 8b.

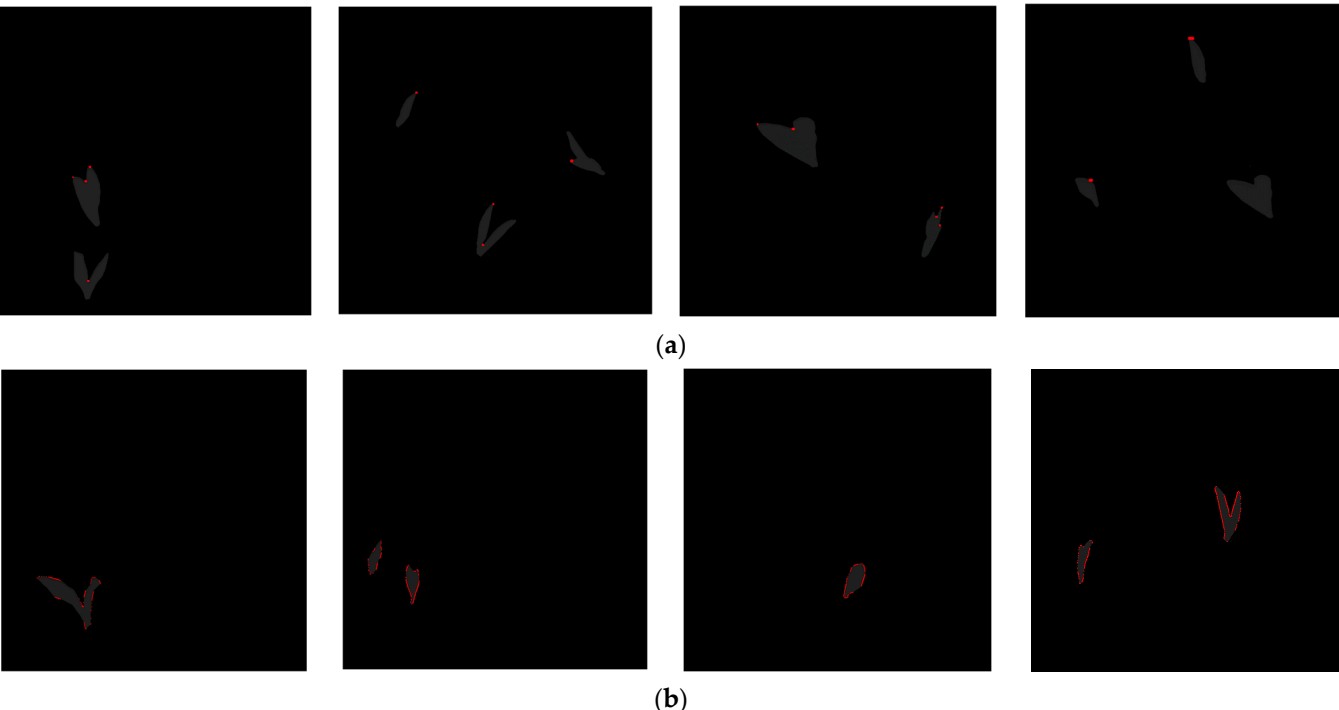

**Figure 8.** A diagram of anomalous number of corner points: (**a**) Low number of corner points; (**b**) Excessive number of corner points.

*3.3. Tea Bud Picking Point Extraction*

Figure 9 shows the process of tea bud segmentation to detect corners. Firstly, the original Figure 9a was segmented by the semantic HRNnet_W18 algorithm to obtain Figure 10. Then, median filtering of a $3 \times 3$ kernel is used to smooth image edges and obtain Figure 9c. Subsequently, through the connected domain algorithm, the tea buds that are too small and seriously missing information are filtered out. Meanwhile, the length, width, and starting point coordinates of each tea bud image are saved. Finally, the corner points of each tea bud in Figure 9d were calculated, and the image coordinates of each corner point were saved to obtain Figure 9e.

Due to the different orientations of tea buds in the image, they are roughly divided into four situations: up, down, left, and right, and then the picking point of tea buds is further extracted. It can be seen from the observation that the tea bud from the terminal bud to the tea stalk is generally in the shape of a large at the top and a small at the bottom. According to this objective law, taking the centroid of the tea bud as the starting point and connecting the center of the tea bud's minimum external rectangle, the point farthest from the centroid in this direction is the picking point. Similarly, the opposite direction is the approximate orientation of the tea buds in the image. Tea stalks are roughly oriented and

extracted from picking points, as shown in Figure 10. The red dot, blue dot, green dot, yellow line, and purple dot in the figure, respectively, represent the detected corner point, centroid, image center, tea stem orientation, and final picking point.

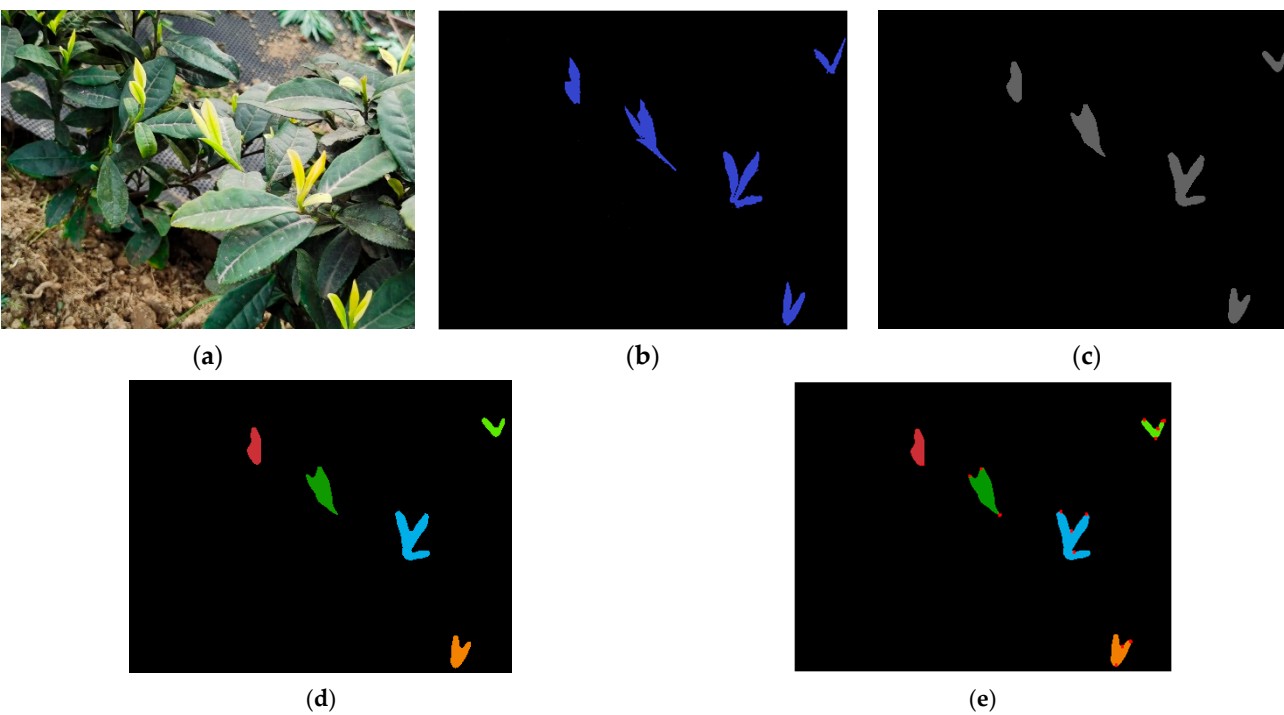

(a)　　　　　　　　　(b)　　　　　　　　　(c)

(d)　　　　　　　　　　　　　　　(e)

**Figure 9.** Tea bud segmentation and processing: (**a**) The original image; (**b**) HRNnet_W18 segmentation; (**c**) 3 × 3 median filtering; (**d**) Connected domain algorithm; and (**e**) Corner detection.

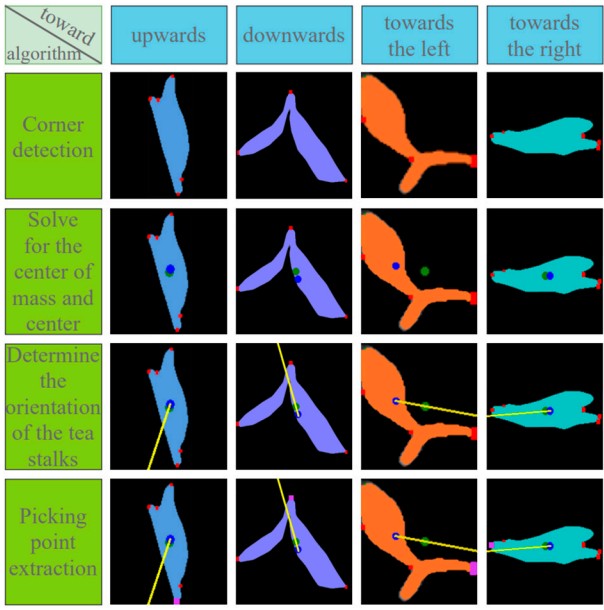

**Figure 10.** The extraction process of picking points in different towards.

### 3.4. Prediction and Analysis of Tea Bud Picking Point

In order to verify whether the method designed in this paper is scientific and effective, a test set of tea bud samples is used to predict picking points. By connecting the center of the minimum external rectangle and the centroid of the tea bud image solved, pointing and connecting the extension line with the corner points were calculated. When the selected

corner meets the requirements of the picking point, it is judged as correct, while the stem that deviates from the picking area is marked as failed. Dividing the correct quantity by the total is determined as the forecast correct ratio. A total of 101 tea buds were tested this time, and according to the top orientation of the tea buds, the data is divided into four types as shown in Table 1. Among them, the ratio of upward tea buds is the largest, which is determined by the upward natural growth posture of tea buds. The downward proportion of tea buds is the least, and the different directions of tea buds in the image are caused by the camera's shooting angle and growth status.

**Table 1.** Prediction of tea bud picking point.

| Buds Towards | Upwards | Downwards | Towards the Left | Towards the Right | Total Value or Average Value |
|---|---|---|---|---|---|
| The total number of buds | 35 | 13 | 30 | 23 | 101 |
| Correct number of picking points predicted | 20 | 8 | 16 | 13 | 56 |
| Forecast correct ratio | 57% | 61% | 53% | 56% | 57% |

After statistical analysis, the forecast-correct ratio for downwards is the highest at 61%, and for leftwards it is the lowest at 53%. The average prediction accuracy for picking points is 57%. In the four directions of tea buds, there is no significant difference in the correct prediction rate for all types. However, to the left and right, there are more pixels in the lower part of the tea bud that cause corner clustering, which leads to misjudgment of direction prediction.

## 4. Discussion

Table 2 studies different picking point location methods applied to tea. It compares the task, datasets, and methods and lists the advantages and disadvantages of each to better explain the experimental results obtained in this paper.

**Table 2.** Studies on different picking point location methods applied to tea.

| No | Year | Reference | Task | Dataset | Methods | Pros and Cons |
|---|---|---|---|---|---|---|
| 1 | 2022 | Long et al. [14] | Identification of tea buds and location of picking points | Shoot from the tea garden using vivo x6s mobile phone | 2 × G-B-R, Edge detection, skeleton extraction | The picking point location using super-green feature and skeleton extraction has strong robustness. |
| 2 | 2022 | Fang et al. [15] | Identification of tea buds and location of picking points | IntelD415, Nikon camera, smartphone; 5000 photos from tea garden base in Huangshan City, China | RGB-HSV, YOLOV4; uses binary contours to extract the image and uses median filtering to remove noise, and then carries out the erosion operation, using the fuse line as the picking point. | The effect is better for small targets, but it is necessary to detect the target image clearly and intact, with clear picking stalk leaves exposed. |
| 3 | 2020 | Chen et al. [16] | Location of tea bud picking point | Panasonic Lumix DC-GF9 and Canon M3. 150 photos from the tea garden | Fast R-CNN, FCN, Opencv | The average accuracy can reach 84.91%, but the occlusion image has a great influence on the recognition effect. |
| 4 | 2021 | Wang et al. [17] | Tea picking spot detection and location | Smartphones; 1400 from a tea garden in Huangshan City, China | Mask R-CNN | The bud rod is identified and segmented and the centroid is selected as the picking point to achieve the optimal effect at a specific Angle |

**Table 2.** *Cont.*

| No | Year | Reference | Task | Dataset | Methods | Pros and Cons |
|---|---|---|---|---|---|---|
| 5 | 2022 | Yan et al. [18] | Tea bud identification and picking location | Iphone;464 from the tea garden in Hangzhou, China | Mask R-CNN | The mAP value of bud point recognition is 0.449, the F2 value is 0.313, but the accuracy of picking point location is 0.949, and the recall rate is 0.910. |
| 6 | 2022 | Yan et al. [19] | Image Segmentation of Tea Bud and coordinate location of picking Point | Redmi Note 7Pro mobile phone; 1110 from tea gardens in Guizhou, China | DeeplabV3+, RGB color separation, Shi-Tomasi algorithm | The tea buds with too many corners and large angles can not be solved by identifying the corners of the connected domain of the Shii-Tomasi algorithm. |
| 7 | 2022 | Qi et al. [20] | Identification of litchi picking point | 200 from the Internet. | PSPNet, YOLOv5, binarized and skeletonized | The picking order is determined by setting the dividing line, so that the picking point is fault-tolerant, and the advantages of PSPnet over U-Net and Deeplab_v3+ are determined. |
| 8 | 2022 | Li et al. [21] | Identification and picking Point location of Longan of UAV | 810 images from Longan Germplasm Resources Nursery, Guangdong Academy of Agricultural Sciences, China | Improved yolov5s, improved DeepLabv3+ | The connection line is determined by pixel distance to fit the actual axis and picking point of fruit branch, which takes 0.58 s to meet the picking requirements, and the environmental anti-interference ability needs to be improved. |
| 9 | | This study | Tea bud identification and picking location | Smart phones; 200 from Emei Mountain Tea Garden in China | HRNet_W18, U-Net, Fast-SCNN, Deeplabv3+ | HRNet_W18 has better segmentation effect and robustness, and this method can meet the target point detection of different angles at the same time. |

In this experiment, we found that due to the complex lighting conditions in the natural environment and the diversity of biological characteristics of tea buds, the identification of tea picking points is difficult. According to the significant color difference between tea buds and old leaves, the use of threshold segmentation based on ultra-green features [14] can effectively extract foreground information. Fang et al. [15] use RGB-HSV color transformation to obtain the outline of the extracted bud leaves and locate the picking point based on the morphological algorithm, but the traditional color separation image method is easily affected by the weather and other natural environments. For example, because the surface of young tea leaves is covered with a fine layer of villi, under direct sunlight on a sunny day, tea buds will cause reflection and affect the effect of segmentation and detection, as shown in Figure 11.

Compared with traditional methods, deep learning often directly uses trained algorithms for direct image segmentation, such as Fast R-CNN [16], Mask R-CNN [17,18], Improved DeepLabv3+ [19], etc. Due to the use of datasets from different times and environments for training, deep learning-based segmentation algorithms perform better in natural environments than traditional segmentation algorithms. After image segmentation, Chen et al. [16] use the full convolutional network (FCN) to determine the area to be picked and then use the centroid method to determine the picking point. Wang et al. [17] use the semantic segmentation function of the model to directly segment the picking area after classifying the leaves, but the shapes and colors of the stems of the whole tea plant are very similar, which leads to erroneous detection results. Yan et al. [18] After determining the shot's main mask, use the minimum circumscribed rectangle to directly obtain the 2d coordinate information of the picking point. Yan et al. [19] used the Shi Tomasi algorithm

to calculate the corners of the maximum connected area. At the same time, the corner point with the lowest vertical coordinate is identified as the corresponding pick point. However, these algorithms are all tested at fixed shooting angles and may miss the detection of tea buds in non-standard growth states.

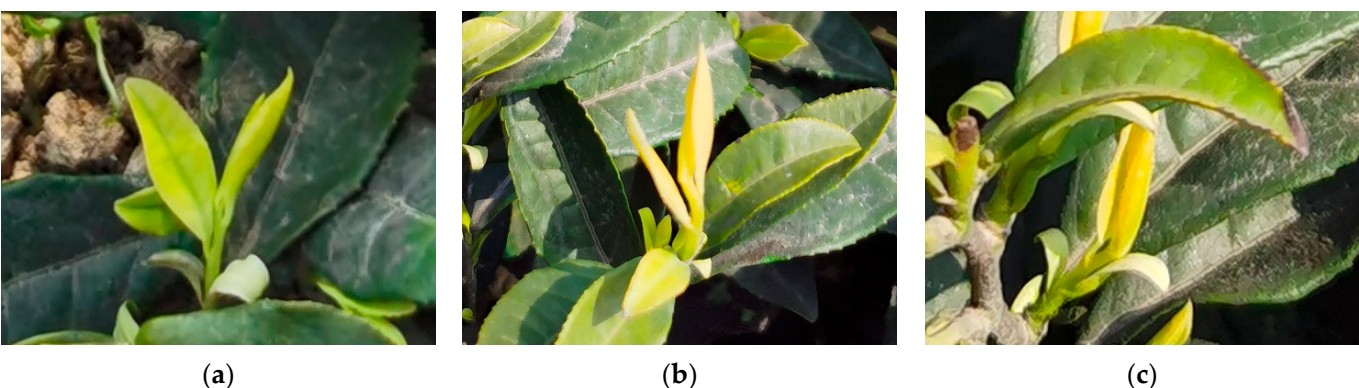

(**a**)                                             (**b**)                                             (**c**)

**Figure 11.** Different states of tea buds: (**a**) backlight; (**b**) direct sunlight; (**c**) occlusion.

Qi et al. [20] look for the picking point of litchi, use PSPNet to segment the fruit stalk, binarize and then skeletonize the segmented image of the fruit stalk, and find the picking point on the skeleton by looking for the centerline. Li et al. [21] use a UAV equipped with an RGB-D camera to find longan picking points. The segmented fruit stalks are fitted with different pixel connection lines and central axes to get the main fruit branches, select the picking points, and return to deep space coordinates. It can be confirmed that image segmentation and corner points can provide advance assistance for selecting pick points.

In our research, we first compared the four methods of HRNet_W18, U-Net, Fast-SCNN, and Deeplabv3+ and selected HRNet_W18 as the best test method. Through binarization and corner selection, combined with the centroid and image center extension line to determine picking points, we made this method have better reliability and robustness and can adapt to the non-traditional tea bud picking point recognition with a fixed angle and shooting height.

## 5. Conclusions

In this study, we propose a new tea bud picking point recognition method to identify tea buds with different poses and directions. Due to the biological characteristics of tea buds, there are significant differences in their growth positions. Picking points not directly below them will cause difficulties in identification. The method is mainly divided into two parts: image segmentation and picking point localization. To select a semantic segmentation algorithm with better performance, four algorithm models (Deeplabv3+, U_Net, HRNet_W18, and FastSCNN) were used for the tea bud segmentation test. The *mIoU* are 0.7859, 0.7964, 0.81, and 0.748, respectively. Then, based on the HRNet_w18 model segmentation images, a $3 \times 3$ filter kernel with the least number of abnormal tea buds was selected, the corner points of tea buds and the orientation of tea stalks were detected in parallel, and finally, the picking points of tea buds were obtained. The test set was divided into four categories according to the orientation of the tea stalks. Among the 101 targets in four, the average prediction accuracy of picking points was 57%, which proved the scientific and effective method. The method promoted in this paper combines semantic segmentation and tea stem orientation and has been tested and has good performance, which can provide a reference for machine-picking tea buds in the future.

The established cloud platform can not only share data but also calculate tea bud coordinates in the cloud. This will greatly reduce the energy consumption and purchase cost of the machine. The bidirectional feedback of cloud platform data is better completed, replacing the ultra-high requirements of airborne processors, and at the same time, it is convenient for developers to provide users with algorithm update iterations. In the application of agriculture, the advantages of the cloud platform will greatly change the working mode of agriculture and make unmanned farms a possibility.

**Author Contributions:** Conceptualization, J.L.; formal analysis, Z.G.; methodology, Q.S.; software, Z.Y.; validation, Q.S. and Z.Y.; writing—original draft, J.L. and Z.Y.; writing—review and editing, W.M. and J.L. All authors have read and agreed to the published version of the manuscript.

**Funding:** The research was funded by the Chengdu Science and Technology Bureau (2022YF0501127SN) and the Department of Science and Technology of Sichuan Province (2022YFG0147).

**Data Availability Statement:** Not applicable.

**Conflicts of Interest:** The authors declare no conflict of interest.

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
