# Peer review of "A Machine Vision-Based Method for Tea Buds Segmentation and Picking Point Location Used on a Cloud Platform"

_agronomy, doi:10.3390/agronomy13061537_

Round 1

Reviewer 1 Report

Detailed notes:

(1) Keywords that repeat the title should be avoided.

(2) Please standardize the way of citing bibliography items in accordance with editorial recommendations.

(3) The space character is missing in many places in the text. Please remove the double spaces.

(4) The multiplication sign (×) is used in texts instead of the word na or times between numerical numbers. It is then folded with a gap on both sides. Do not replace it with the letter x and the * sign.

(5) Figure 1 should be deleted - it does not provide relevant information.

(6) We do not put a dot after the word Figure. Please correct in the text.

(7) Incorrect format in formula (2).

(8) The last paragraph on page 3 is in the wrong font size.

(9) In Table 2, increase the width of the Reference column so that the year in parentheses fits on one line.

(10) Conclusions must take a more synthetic form and result directly from the analyzes carried out. Repetitions regarding the course and scope of the work carried out should be removed.

Author Response

Dear reviewer,

We would like to thank you for the time and effort spent in reviewing the manuscript. These comments and professional advice help to improve academic rigor of our article. Based on your suggestion and request, we have made corrected modifications on the revised manuscript. Please see the attachment.

Reviewer 2 Report

Dear Authors,

Thank you for submitting your work to Agronomy. Robotizing tea bud picking is indeed a significant challenge, and if realized, it could have a significant impact on the tea industry. I have read your paper, and while I can see that you utilized HRNet to segment tea buds and designed an algorithm to identify the picking point, I have some comments that need to be addressed.

  1. Lines 309 to 360 appear to be a literature review. As such, it would be more appropriate to integrate this section into the introduction or literature review section of your paper, rather than in the discussion section.

  2. You should provide a detailed description of your dataset, including how the data was collected, under what conditions, and whether the conditions varied enough to ensure the generalizability of the model. Additionally, please provide information about the split ratio between train, test, and validation data.

  3. Your method consists of two parts: tea bud segmentation and picking point selection. While you have used mIoU as a quantitative metric for segmentation, you have not provided information about how you evaluated picking point selection quantitatively.

  4. In line 339, you stated that other methods are slow. To support this claim, you should add an inferencing time comparison for your method and others.

  5. You mentioned using a median filter after segmentation. However, it is not clear why this step is necessary. From Figure 10, it is evident that applying a median filter will harm the sharpness and IoU of the segmentation results, which you seem to care about in line 217, but after applying a median filter, all network differences may diminish or even disappear.

  6. I do not see the necessity of adding a cloud platform to the paper. The cloud platform does not seem to contribute to the main theme of the paper and may even detract from it. While you state that the cloud platform can be used for real-time segmentation, your cloud interface appears to be more like a labeling and human-use interface. Additionally, there is no analysis of the latency, bandwidth, network reliability, etc., to prove that this platform can be used for real-time segmentation.

I hope you find my comments helpful, and I look forward to seeing the revised version of your paper.

Best regards,

Author Response

(The authors gave the same response as above.)

Reviewer 3 Report

The authors proposed a tea bud segmentation and picking point location method based on a high-resolution network model (HRNet_W18). The idea is interesting and the paper is well writen with some minor errors, but the authors need to pay careful attention to some important topics regarding their proposal and comparison to other work, as listed in sequence. Please note that the comments are intented to improve paper quality and readers' understanding.

The main part of the paper can be considered to be Table 2. It describes related work that also deal with tea bud segmentation and picking point location estimation, providing the dataset used, methods and pros and cons. What happens to be the most interesting part of the paper is also its weakest point. For the paper to be accepted for publication, the authors must make clear how their approach compares to the other methods. Table 2 is not enough to satisfy such comparison, as the authors only provide descriptions regarding each of the related work, and do not compare their work with the others, quantitatively or qualitatively. In fact, reading the content of Table 2 gives the impression that the results obtained by the authors are not better than the ones from the state of the art.

The authors must clearly explain the strong points of their algorithm and also its limitations, pointing out how it advances the state of the art. There is no point in publishing a paper that does not bring any advantage (at least in one of these topics: accuracy, performance, automation, etc.) compared to the state of the art. In summary, my suggestion to the authors is that they make clear how the proposed work surpasses all aforementioned works and what is the novelty it brings.

More general comments and minor errors are listed as follows.

"So developing" -> "So, developing"

"Xuemei et al. [3]proposed" -> "Xuemei et al. [3] proposed"

" Chen et al. [4]performed" -> " Chen et al. [4] performed"

"Thangavel et al. [5]proposed" -> "Thangavel et al. [5] proposed"

"Chen et al. [7]used" -> "Chen et al. [7] used"

"Zhang et al. [8]implemented" -> "Zhang et al. [8] implemented"

"Chen et al. [9]used" -> "Chen et al. [9] used"

"Zhang et al. [12]used " -> "Zhang et al. [12] used "

"Franco et al. [13]Real-time monitoring of temperature and humidity parameters during seed germination based on Cloudino-iot and FIWARE platforms." -> please rewrite

" 16G," -> " 16GB,"

"Figure. 1." -> "Figure 1."

"In this study; four semantic segmentation algorithms; each with its own characteristics; U-Net; High-Resolution Network (HRNet_W18); Fast Semantic Segmentation Network (Fast-SCNN); and Deeplabv3+, were selected for processing images. U-net has a symmetric network structure and uses its low-level features for feature fusion; which is mainly applied to seamless segmentation tasks. However; this network structure will reduce the resolution of output results. Fast-SCNN in multibranch network structure; branches sharing shallow network can simplify the whole network structure and reduce the amount of computation. Moreover; the learning to downsample module is designed in this network; which can be used by branches to extract low-level features. Based on the learning to downsample module and two branches; the real-time semantic segmentation network is constructed. Deeplabv3+ proposes a new encoder-decoder architecture that focuses on target boundary information to improve segmentation while using the Xception module and applying depth-wise separable convolution into the Atrous Spatial Pyramid Pooling (ASPP) and decoder modules to improve the computational speed of the network" -> "In this study, four semantic segmentation algorithms, each with its own characteristics (U-Net, High-Resolution Network (HRNet_W18), Fast Semantic Segmentation Network (Fast-SCNN) and Deeplabv3+), were selected for processing images. U-net has a symmetric network structure and uses its low-level features for feature fusion, which is mainly applied to seamless segmentation tasks. However, this network structure will reduce the resolution of output results. Fast-SCNN works in a multibranch network structure: branches sharing shallow network can simplify the whole network structure and reduce the amount of computation. Moreover, the learning to downsample module is designed in this network, which can be used by branches to extract low-level features. Based on the learning to downsample module and two branches, the real-time semantic segmentation network is constructed. Deeplabv3+ proposes a new encoder-decoder architecture that focuses on target boundary information to improve segmentation while using the Xception module and applying depth-wise separable convolution into the Atrous Spatial Pyramid Pooling (ASPP) and decoder modules to improve the computational speed of the network"

" 7 open source" -> what? please explain

"Figure. 2." -> "Figure 2."

" Figure. 3." -> " Figure 3."

"Figure. 4." -> "Figure 4."

"Python3.7.5" -> "Python 3.7.5"

"Python3.7.5" -> "Python 3.7.5"

" Figure. 7 " -> " Figure 7 "

" Figure. 8," -> " Figure 8,"

"Figure. 9" -> "Figure 9"

" Figure. 9 (a)." -> " Figure 9 (a)."

"Figure. 9 (b)." -> "Figure 9 (b)."

"Figure. 10" -> "Figure 10"

"Figure. 10 (a)" -> "Figure 10 (a)"

"Figure. 10 (b)." -> "Figure 10 (b)."

"Figure. 10 (c). " -> "Figure 10 (c). "

"Figure. 10 (d)" -> "Figure 10 (d)"

"Figure. 10 (e)." -> "Figure 10 (e)."

"Figure. 12 (a)" -> "Figure 12 (a)"

"Figure . 12 (b)" -> "Figure 12 (b)"

"Figure . 12 (c)" -> "Figure 12 (c)"

" Figure . 12 (d)" -> " Figure 12 (d)"

"Table 2 compares recent studies on various determine the picking point methods applied to tea. " -> please rewrite

"Table 2. Comparison of studies on various determine the picking point methods applied to tea" -> please rewrite

"Qi et al.[20] looks" -> "Qi et al.[20] look"

" first uses" -> " first use"

"then uses" -> "then use"

"binarizes and then skeletons" -> "binarize and then skeletonize"

"and finds" -> "and find"

"Li et al.[21] uses" -> "Li et al.[21] use"

"This this study" -> "This study"

Author Response

(The authors gave the same response as above.)

Round 2

Reviewer 2 Report

I believe the manuscript has been sufficiently improved to warrant publication in Agronomy. 

Author Response

Thank you very much for improving the manuscript.

Reviewer 3 Report

Thank you for updating the text considering most of the aforementioned comments, but there is still room for improvement.

Regarding the main issue of the paper (about the comparison with other works from the state of the art), it is still not clear how the proposed method differs and surpasses what already exists. It is necessary to perform a more quantitative comparison to show how better the proposed solution performs. It is not sufficient to list in text the differences of the approaches. Added to that, it is important to, for instance, use the same datasets previous works used to validate their work and run the proposed solution on these datasets. This way, the authors will be able to directly compare the works. Another alternative (if you have the code available from previous works) is to run the state of the art code on your own dataset. Anyway, this comparison is mandatory for the publication to be accepted.

More general comments and minor errors found are listed as follows.

"Table 2 studies" -> "Table 2 compares studies"

"In this experiment, we found that Due" -> "In this experiment, we found that due"

"leaves, The use" -> "leaves, the use"

"Xu et al.[15] uses" -> "Xu et al.[15] use"

"and locates the" -> "and locate the"

"environment, For example," -> "environment. For example,"

"Due to use datasets of different times" -> please rewrite

"Chen et al.[16] uses" -> "Chen et al.[16] use"

"Wang et al.[17] uses" -> "Wang et al.[17] use"

"Yan et al.[18] After" -> "Yan et al.[18], after"

"uses the minimum" -> "use the minimum"

"2d coordinate" -> "2D coordinate"

"uses PSPNet" -> "use PSPNet"

" selection, Combining" -> " selection, combining"

"not directly below " -> below what?

"for select a" -> "For selecting a"

"the mIoU" -> "The mIoU"

" images," -> " images, a"

" Among the 101 targets in four," -> please rewrite

Author Response

Dear reviewer,

Thank you for your comments concerning our manuscript entitled “A machine vision-based method for tea buds segmentation and picking point location used on a cloud platform”. Those comments are all valuable and very helpful for revising and improving our manuscript, as well as the important guiding significance to our researches. We have studied comments carefully and have made correction which we hope meet with approval. The main corrections in the paper and the responds to the reviewer's comments are in the attached files.
